# Perception of Synchronized Online Teaching Using Blackboard Collaborate among Undergraduate Dental Students in Saudi Arabia

**DOI:** 10.3390/ijerph191912825

**Published:** 2022-10-07

**Authors:** Abdul Ahad Khan, Chidozie Ifechi Onwuka, Shahabe Saquib Abullais, Nasser Mohammad Alqahtani, Mohammad Zahir Kota, Abosofyan Salih Atta, Shahi Jahan Shah, Mohammed Ibrahim, Shaik Mohammed Asif, Mohamed Fadul A. Elagib

**Affiliations:** 1Department of Oral and Maxillofacial Surgery, King Khalid University, College of Dentistry, Abha 62529, Saudi Arabia; 2Department of Periodontics and Community Dental Sciences, College of Dentistry, King Khalid University, Abha 62529, Saudi Arabia; 3Department of Prosthetic Dentistry, College of Dentistry, King Khalid University, Abha 62529, Saudi Arabia; 4Department of Diagnostic Sciences and Oral Biology, King Khalid, University College of Dentistry, Abha 62529, Saudi Arabia

**Keywords:** online, blackboard, collaborate, perception

## Abstract

Blackboard is a collaborative virtual learning tool used for higher learning that has been found to be an effective and efficient means of interactions between teachers and students and offers effective educational information management. The aim of this research work is to assess the preclinical and clinical dental students’ perception of Blackboard Collaborate as a quality teaching and learning tool as well as to find out areas that might appear as barriers to quality teaching and learning. This cross-sectional study was conducted online using survey monkey involving 245 dental students who had participated in the virtual classroom lectures during the pandemic with 18 students not completing the survey. The survey instrument was a nine-item questionnaire that included the age, sex, and year of study of the students as well as previous exposure to online lectures. The data collated was analyzed using IBM Statistical Package for the Social Sciences (SPSS) Statistics for windows version 22. Among 245 respondents that were enrolled in the study, 227 respondents completed the survey, of which 58.1% (n = 132) were male while 41.9% (n = 95) were females. Of the 227 respondents that completed this study, 74.8% (n = 170) of them experienced minimum to moderate technical problems regarding connectivity during the online sessions while 1.8% (n = 4) of the respondents experienced very severe technical problems. The majority of the respondents 54.2% (n = 123) support the continuation of online lectures even after the pandemic. In conclusion, we found a positive perception of our respondents to online lectures using Blackboard Collaborate. Internet connectivity as well as a decline in the comprehension of the lectures as compared to face-to-face learning were found as barriers to online learning.

## 1. Introduction

Advancements in information technology have led to changes in the mode of educational learning with a tremendous impact on the teaching methods in our educational system. Virtual learning through means of podcasts, messengers, Zoom, discussion boards as well as Blackboard has made interactions between the students and the lecturers as effective as the traditional learning method [1]. Online learning can be of two types: synchronous type, where students are required to join classes at a particular time of the week, and asynchronous type, where students view instructional materials at any time of the week [2]. Blackboard is one of the collaborative virtual learning tools use for higher learning that has been found to be an effective and efficient means of interactions between teachers and students and offers effective educational information management [3]. Additionally, it has been noted to be cost-effective in terms of knowledge reproduction, resulting in an efficient educational management system [3]. The current COVID-19 pandemic made the use of virtual learning via Blackboard and others a solution. However, the use of virtual learning methods such as Blackboard has been tagged in some quarters as digital myopia [4] based on the fact that they are focused on mechanical information rather than on an innovative pedagogical approach to learning [5,6]. Additionally, the success of online learning is dependent on students’ understanding of this mode of communication and their ability to efficiently utilize available digital resource materials [7]. Despite these shortcomings, the use of online learning became imperative during the COVID-19 period, as measures such as social distancing and isolation rules came into place, in order to curtail the spread of the virus thus ensuring the safety and wellbeing of the students and faculty members [8]. Online teachings are tailored to be student-centered and have had a positive impact on medical education before the advent of the COVID-19 pandemic [8,9,10,11]. However, the undergraduate curriculum for the medical and dental faculties entails the acquisition of clinical skills, especially during clinical classes. This requirement was a sort of limitation to the sole use of online teaching since clinical skills competency cannot be impacted online [12]. Furthermore, the COVID-19 pandemic forced universities to be closed; thus, students were not able to attend their routine classrooms for the completion of their course curriculum. Various tasks that needed to be completed including lecture sessions, assignments and clinical tasks, without which the students cannot be graded in their continuous and final assessments were suspended. This necessitated the use of online lecture tools in order to ameliorate the impact of the pandemic on the student’s curriculum even though clinical tasks were not included in the system because of feasibility. However, if the comprehension of the students through the online mode is not adequate, they will not be able to carry out the various assignments to their fullest ability, resulting in poor grades and doctors with less clinical skills. Thus, an assessment of their understanding is very important as it will serve as a guide toward any modifications that can be made in the current teaching modality to suit the situation. In addition, when the students are present in the classroom before the faculty members, they can easily make inquiries regarding the content of the lectures coupled with the fact that the body language and communication skills during the didactic lecture sessions are very beneficial in making the students understand difficult and complex concepts. The above scenario is mostly lacking in online lecture sessions. Hence, there is a need to know the students’ views on the advantages, disadvantages and difficulties they faced while attending online sessions during the pandemic so that measures can be taken for their elimination/reduction and to improve students’ experience. The aim of this study, therefore, is to assess undergraduate dental students’ perception of online teaching and learning as well as to find out areas that might appear as barriers to quality online teaching and learning in Saudi Arabia.

## 2. Materials and Methods

This was a cross-sectional study conducted online using SurveyMonkey, involving 245 dental students who had participated in virtual classroom lectures during the pandemic, with 18 students not completing the survey. The respondents were from 3rd (levels 5 and 6) year to 6th year (levels 11 and 12) dental undergraduates of the College of Dentistry, King Khalid University, Abha, Saudi Arabia.

### 2.1. Survey Instrument

This was a nine-item questionnaire developed based on certain criteria that included students’ excuses for their lecture absenteeism, quiz performances, their expectations and feedback from other faculty members. The questionnaire included the age, sex, and year of study of the students as well as previous exposure to online lectures. Five of the nine-item questionnaire were related to session duration, technical challenges, lecture feedback, lecture delivery and students’ understanding of the topic with 2–5 sub-items. The questionnaire was validated through a pilot study involving 30 students that were not part of the final study.

### 2.2. Ethical Consideration

Participation was voluntary and consent was given by all respondents by ticking the mandatory consent section in the survey monkey. Survey Monkey was shared through various WhatsApp groups and can only be accessed through student emails, ensuring that students complete the survey only once. Ethical clearance was applied for and approved by the College of Dentistry Institutional Review Board (IRB).

### 2.3. Data Analysis

The data collated was analyzed using IBM Statistical Package for the Social Sciences (SPSS) Statistics for windows version 22.0 (IBM Corp., Armonk, NY, USA). Categorical variables were presented in percentages (%), while continuous variables were presented as means and standard variation (SD). The composite scores of issues related to technical sessions and the understanding of the topic were calculated using the weighted average method. The inter-group statistical comparison of the distribution of categorical variables was tested using the Chi-Square test or Fisher’s exact probability test where more than 20% of cells have an expected frequency of less than five. The inter-group statistical comparison of the distribution of means of composite scores was tested using an independent sample t test for two groups and analysis of variance (ANOVA) for more than two groups. The underlying normality assumption was tested before subjecting the composite scores to the t test and ANOVA. Statistical significance was inferred at *p*-values < 0.05. All the hypotheses were formulated using two-tailed alternatives against each null hypothesis.

### 2.4. Results

Among the 245 respondents who were enrolled in this study, 227 respondents completed the survey, of which 58.1% (n = 132) were male, while 41.9% (n = 95) were females. Year 3 students (levels 5 and 6) accounted for 27.3% (n = 62) of the respondents, while 22% (n = 50) of the respondents were in 6th year (levels 11 and 12). Of the 227 respondents, 72.2% (n = 164) had been exposed to online lectures before the advent of the COVID-19 pandemic majorly through Blackboard Collaborate, as seen in Table 1.

Most of the respondents had online lecture sessions of a duration of 45 min and above, while 35.7% (n = 81) had 60 min as the maximum duration. The majority of respondents 49.8% (n = 113) opted for 45 min as the maximum session duration for a topic in order to ensure good receptivity. The ideal free time intervals between lectures were considered by 35.7% (n = 81) of respondents to be between 11–15 min, while 29.5% (n = 67) opted for a 6–10-minute free time interval, as seen in Table 2.

Of the 227 respondents in this study, 74.8% (n = 170) experienced minimum to moderate technical problems regarding connectivity during online sessions, while 1.8% (n = 4) of the respondents experienced very severe technical problems. The majority of the respondents 52.9% (n = 120) were neutral on the timing of the online lecture, while 37.8% found the timing of the online lectures they attended convenient. One hundred and seventy-eight respondents (77.6%) found the learning experience with online sessions from home comfortable as compared to face-to-face sessions, even though half of this group did not have a better understanding of the lectures as compared to face-to-face lectures. The majority of the respondents 54.2% (n = 123) support the continuation of online lectures, even after the pandemic, as seen in Table 3.

As regards topic delivery and understanding of the topics, 64.8% (n = 147) were of the opinion that online lectures should be made more interactive, with 26.4% (n = 60) respondents finding it difficult to understand the lecture concepts through the online sessions as compared to face-to-face sessions—even though the majority of 227 respondents 47.6% (n = 108) were neutral. One hundred and seven respondents (47.1%) agreed that the visibility of faculty members affects their understanding of the lecture topic, as seen in Table 4.

The distribution of responses and mean score regarding the comfort of the learning experience with online sessions from home as compared to face-to-face sessions differs significantly between four different groups of levels (years) of respondents (*p*-value < 0.05), with respondents in lower classes being more comfortable with online lectures compared to higher classes (*p* = 0.002, 0.013), as seen in Table 5.

The distribution of the responses regarding the ease/difficulty to understand the lecture concepts through the online sessions as compared to the face-to-face sessions differs significantly between the four different groups of levels (years) of respondents (*p* = 0.003), with more respondents in lower levels (year) finding it more difficult to understand the lecture concept through an online session than higher level; however, there was no significant difference in mean score across the levels in that regard (*p* = 0.138) as seen in Table 6.

More males found the timing of online lectures very convenient as compared to females; however, the majority of the respondents were neutral concerning the online session timing. (*p* = 0.001) as seen in Table 7.

The distribution of the responses regarding the ease to clarify their doubts as relates to the lecture topic through Blackboard Collaborate differs significantly between male and female respondents, with males finding it easier (*p*-value < 0.05). However, their responses regarding the ease/difficulty to understand the lecture concepts through the online sessions as compared to face-to-face sessions differ significantly between male and female respondents, with more males finding it difficult when compared to females (*p* = 0.001), as seen in Table 8.

## 3. Discussion

The COVID-19 pandemic necessitated schools to move their teaching to an online platform in order to mitigate the impact of the pandemic on students’ academic lives. Educational institutions, including dental schools, were left with no other choice than to close in-person school activities as social distancing and other rules came into force as the world battled the then poorly understood viral pandemic. As governments gradually relaxed rules and in-person schooling gradually resumed, it became necessary to evaluate online classes with regard to students’ perceptions, especially among dental students, in order to tailor them according to their expectations for future uses. Our study found that the longest online lecture attended by the majority of respondents was between 45–60 mins, while a lecture duration lasting up to 90 mins was the least preferable to respondents. However, most students preferred a lecture period between 30–45 mins in order to ensure maximum attention and receptivity. Studies have shown learning as well as teaching are dependent on students’ attention [13,14,15]. Students’ peak attention period usual lasts 10–15 mins, after which it declines; thus, longer lecture durations will affect students’ receptivity, as our study found more students prefer 45 mins or less [13,16,17]. Bradbury in his study, however, disputes the short attention period studies, noting that the 10–15-mins attention time period is frothed with errors [18]. The majority of the students in our study also preferred a time interval between lectures to be between 6 mins to 15 mins in order to be refreshed and restore attention on the next lecture as suggested in Eze and Misava‘s study [19]. Technical issues in relation to connectivity as well learning from home were found to have an effect on as online learning, even though only 1.8% of our study respondents recorded it as severe and half of respondents that found online lectures from home comfortable noted that it affected their concentration. Our findings differ on technical issues but were similar on family distractions, with studies by Dost et al. [20] and Khali et al. [21] that found connectivity issues and family distractions as the main factors that affect online learning. Contrary to our findings and in another Saudi study [22], 80% of respondents in Yunfei et al.’s [23] study found online lectures uncomfortable. Previous exposure to online lectures by the majority of our respondents may have accounted for the differences in connectivity findings between the studies. The timing of online lectures in our study was considered by our respondents to be convenient even though the majority of them were neutral, this is in contrast to Dost et al. [20], who found the timing of online lectures to be the main barrier in their own study. The difference in our findings may be a result of the fact that our online lectures were held in the evening, when the majority of students would be done with their day’s activities and are with family members. This may also play a role in the lack of concentration reported by half of respondents who found online lectures from home convenient. A larger sample size in Dost et al. [20] study could be the reason for the discrepancies in the findings between our study and theirs. In contrast to Yunfei et al.’s [23] findings, the majority of our study respondents preferred the continuation of online learning—probably due to the fact that it is convenient for them. However, 68.4% want it to be more interactive; this is similar to findings in a Polish study [24], where respondents reported less activity during the online lecture sessions. Studies have noted the gamification (game design elements are used in non-game contexts) of online teaching methods as a means of making the sessions very interactive [25,26]. About half of the respondents in our study were of the opinion that the visibility of faculty members has a positive effect on their understanding of the topics.

### 3.1. Intergroup Comparison

In our study, we found that respondents in lower-level clinical classes were more comfortable but had more difficulty understanding the lectures in online classes as compared to higher classes. This finding is in agreement with a previous notion expressed by Chiu et al. in their editorial, where they noted that fresh students may find it more difficult to understand new concepts as compared to older students [27]. Additionally, students in higher classes were more used to face-to-face classes and were more exposed as compared to those in lower classes; this could also account for the differences between the classes. Furthermore, an absence of a clinical framework may also have played a role in the higher classes finding online learning less comfortable. Contrary to our findings, a Polish study ^24^ on medical students found no significant difference between classes as regards enjoying online lectures as well as no difference between genders. This insignificant finding between genders differs from our findings, where there were significant gender differences. Our findings could be explained by gender theory, which suggests that there are gender differences in events’ assessments, with females more likely to adapt to changes as compared to males [28,29].

In conclusion, we found a positive perception of our respondents to online lectures using Blackboard Collaborate. Internet connectivity issues as well as a decline in comprehension of the lectures as compared to face-to-face learning were found as barriers to online learning.

### 3.2. Recommendations

Efforts should be made to make online learning more interactive by allowing the students to view their lecturers during the sessions. In addition, online lectures should not be longer than 45 mins and a minimum of a 15-minute interval should be allowed between lectures.

### 3.3. Limitations of the Study

This study is limited by the fact that the teaching method was restricted to Blackboard Collaborate only. In addition, the study was based on a single institution’s experience.

### 3.4. Further Research

Multiple institutional perceptions of online learning using different virtual learning tools among dental students should be explored.

## Figures and Tables

**Table 1 ijerph-19-12825-t001:** Distribution of demographic characteristics of the respondents.

Characteristics		No. of Respondents	% of Respondents
Sex	Male	132	58.1
	Female	95	41.9
Age group (years)	18–22	109	48.0
	23–27	118	52.0
Current academic level	Year 3 (Levels 5–6)	62	27.3
	Year 4 (Levels 7–8)	61	26.9
	Year 5 (Levels 9–10)	54	23.8
	Year 6 (Levels 11–12)	50	22.0
Have you ever had online lecture sessions before the COVID-19 pandemic?	Yes	164	72.2
	No	63	27.8
Means of online lecture	Zoom	48	21.1
	Blackboard Collaborate	162	71.4
	Other	14	6.7

**Table 2 ijerph-19-12825-t002:** Distribution of responses related to duration of the session.

Duration of the Session		No. of Respondents	% of Respondents
How many minutes was the longest online lecture session you attended?	30 min	21	9.3
	45 min	56	24.7
	60 min	81	35.7
	75 min	25	11.0
	90 min	44	19.4
How many minutes was the shortest online lecture session you attended?	<10 min	23	10.1
	15 min	49	21.6
	20 min	66	29.1
	25 min	45	19.8
	30 min	44	19.4
What should be the maximum duration of a session for a topic in order to ensure good receptivity?	30 min	74	32.6
	45 min	113	49.8
	60 min	30	13.2
	75 min	6	2.6
	90 min	4	1.8
What should be the free time interval between two sessions for good receptivity?	0–5 min	25	11.0
	6–10 min	67	29.5
	11–15 min	81	35.7
	16–20 min	26	11.5
	21–25 min	28	12.3

**Table 3 ijerph-19-12825-t003:** Distribution of responses related to technical session.

Related to Technical Session		No. of Respondents	% of Respondents
Did you experience any technical problems regarding connectivity during the online sessions that affected your understanding of the lecture?	No problem	41	18.1
	Minimum problem	85	37.4
	Moderate problem	85	37.4
	Severe problem	12	5.3
	Very severe problem	4	1.8
How convenient were the timings of the online sessions you attended?	Very convenient	33	14.5
	Convenient	53	23.3
	Neutral	120	52.9
	Not Convenient	14	6.2
	Discomforting	7	3.1
How beneficial did you find the online sessions, especially concerning the effects on your remembering of the concepts of the lecture?	Very beneficial	41	18.1
	Somewhat beneficial	66	29.1
	Neutral	88	38.8
	Not Beneficial	26	11.5
	Not at all beneficial	6	2.6
How comfortable was your learning experience with online sessions from your home as compared to face-to-face sessions?	Very comfortable and able to concentrate better	88	38.8
	Comfortable and not able to concentrate better	88	38.8
	Not very comfortable but able to concentrate	30	13.2
	Not comfortable and not able to concentrate better	17	7.5
	Very uncomfortable and a waste of time	4	1.8
Please state your degree of support in favor of such online lecture sessions in the future.	Strongly supported	44	19.4
	Supported	79	34.8
	Neutral	83	36.6
	Unsupported	17	7.5
	Strongly unsupported	4	1.8

**Table 4 ijerph-19-12825-t004:** Distribution of responses related to understanding of the topic/delivery of the topic.

Related to Understanding of the Topic/Delivery of the Topic		No. of Respondents	% of Respondents
Did you feel and agree that the session needs to be more interactive?	Strongly agree	52	22.9
	Agree	95	41.9
	Neutral	65	28.6
	Disagree	12	5.3
	Strongly disagree	3	1.3
How easy do you think it would be to clarify your doubts as related to the lecture topic through Blackboard collaborate?	Very easy	45	19.8
	Easy	64	28.2
	Neutral	89	39.2
	Difficult	25	11.0
	Very Difficult	4	1.8
How easy/difficult was it to understand the lecture concepts through the online sessions as compared to the face-to-face sessions?	Very Difficult	17	7.5
	Difficult	43	18.9
	Neutral	108	47.6
	Easy	45	19.8
	Very easy	14	6.2
Do you agree that the visibility of the faculty members affects the understanding of the lectures?	Strongly agree	28	12.3
	Agree	79	34.8
	Neutral	89	39.2
	Disagree	24	10.6
	Strongly disagree	7	3.1
How well did the faculty members explain topics on Blackboard as compared to face-to-face sessions?	Very good	31	13.7
	Good	72	31.7
	Neutral	98	43.2
	Bad	23	10.1
	Very bad	3	1.3
What do you think about the performance of online sessions in relation to audio quality as compared to face-to-face lectures?	Very good	31	13.7
	Good	52	22.9
	Neutral	105	46.3
	Bad	36	15.9
	Very bad	3	1.3
What do you think about the performance of online sessions in relation to the speed of content delivery?	Very fast	24	10.6
	Fast	50	22.0
	Neutral	128	56.4
	Slow	20	8.8
	Very slow	5	2.2

**Table 5 ijerph-19-12825-t005:** Distribution of responses related to technical session according to level of study.

		Year 3 (n = 62)		Year 4 (n = 61)		Year 5 (n = 54)		Year 6 (n = 50)		
Related to Technical Session		n	%	n	%	n	%	n	%	*p*-Value
Did you experience any connectivity-related technical problems during the online sessions that affected your understanding of the lecture?	No problems	10	16.1	9	14.8	12	22.2	10	20.0	0.594 ^NS^
	Minimum problems	18	29.0	23	37.7	20	37.0	24	48.0	
	Moderate problems	30	48.4	23	37.7	19	35.2	13	26.0	
	Severe problems	4	6.5	4	6.6	2	3.7	2	4.0	
	Very severe problems	0	0.0	2	3.3	1	1.9	1	2.0	
	Mean ± SD	2.45 ± 0.84		2.46 ± 0.94		2.26 ± 0.91		2.20 ± 0.88		0.304 ^NS^
How convenient were the timings of the online sessions you attended?	Very convenient	17	27.4	6	9.8	2	3.7	8	16.0	0.007 **
	Convenient	15	24.2	10	16.4	15	27.8	13	26.0	
	Neutral	26	41.9	34	55.7	33	61.1	27	54.0	
	Not Convenient	4	6.5	6	9.8	2	3.7	2	4.0	
	Discomforting	0	0.0	5	8.2	2	3.7	0	0.0	
	Mean ± SD	2.27 ± 0.94		2.90 ± 0.99		2.76 ± 0.75		2.46 ± 0.81		0.001 ***
How beneficial did you find the online sessions, especially concerning the effects on your remembering of the concepts of the lecture?	Very beneficial	17	27.4	10	16.4	2	3.7	12	24.0	0.021 *
	Somewhat beneficial	15	24.2	21	34.4	15	27.8	15	30.0	
	Neutral	25	40.3	16	26.2	29	53.7	18	36.0	
	Not Beneficial	4	6.5	11	18.0	6	11.1	5	10.0	
	Not at all beneficial	1	1.6	3	4.9	2	3.7	0	0.0	
	Mean ± SD	2.31 ± 1.00		2.61 ± 1.11		2.83 ± 0.82		2.32 ± 0.96		0.014 *
How comfortable was your learning experience with online sessions from home as compared to face-to-face sessions?	Very comfortable and able to concentrate better	31	50.0	18	29.5	14	25.9	25	50.0	0.002 **
	Comfortable and not able to concentrate better	23	37.1	27	44.3	26	48.1	12	24.0	
	Not very comfortable but able to concentrate	4	6.5	5	8.2	9	16.7	12	24.0	
	Not comfortable and not able to concentrate better	4	6.5	9	14.8	3	5.6	1	2.0	
	Very uncomfortable and a waste of time	0	0.0	2	3.3	2	3.7	0	0.0	
	Mean ± SD	1.69 ± 0.86		2.18 ± 1.12		2.13 ± 0.99		1.78 ± 0.88		0.013 *
Please state your degree of support in favor of such online lecture sessions in the -future.	Strongly supported	17	27.4	8	13.1	6	11.1	13	26.0	0.078 ^NS^
	Supported	16	25.8	19	31.1	26	48.1	18	36.0	
	Neutral	25	40.3	24	39.3	18	33.3	16	32.0	
	Unsupported	2	3.2	9	14.8	3	5.6	3	6.0	
	Strongly unsupported	2	3.2	1	1.6	1	1.9	0	0.0	
	Mean ± SD	2.29 ± 1.01		2.61 ± 0.95		2.39 ± 0.83		2.18 ± 0.89		0.093 ^NS^

NS: Not significant, SD: Standard deviation. *: Significant. **: More significant. ***: Very significant.

**Table 6 ijerph-19-12825-t006:** Distribution of responses related to understanding of the topic/delivery of the topic according to level of study.

		Year 3 (n = 62)		Year 4 (n = 61)		Year 5 (n = 54)		Year 6 (n = 50)		
Related to Understanding of the Topic/Delivery of the Topic		n	%	n	%	n	%	n	%	*p*-Value
Did you feel and agree that the session need to be more interactive?	Strongly agree	14	22.6	12	19.7	11	20.4	15	30.0	0.935 ^NS^
	Agree	26	41.9	24	39.3	24	44.4	21	42.0	
	Neutral	18	29.0	22	36.1	14	25.9	11	22.0	
	Disagree	3	4.8	3	4.9	4	7.4	2	4.0	
	Strongly disagree	1	1.6	0	0.0	1	1.9	1	2.0	
	Mean ± SD	2.21 ± 0.91		2.26 ± 0.83		2.26 ± 0.93		2.06 ± 0.93		0.630 ^NS^
How easy do you think it could be to clarify your doubts as related to the lecture topic through Blackboard collaborate?	Very easy	18	29.0	11	18.0	8	14.8	8	16.0	0.008 **
	Easy	17	27.4	9	14.8	22	40.7	16	32.0	
	Neutral	21	33.9	29	47.5	22	40.7	17	34.0	
	Difficult	5	8.1	12	19.7	0	0.0	8	16.0	
	Very Difficult	1	1.6	0	0.0	2	3.7	1	2.0	
	Mean ± SD	2.26 ± 1.02		2.69 ± 0.99		2.37 ± 0.87		2.56 ± 1.01		0.078 ^NS^
How easy/difficult was it to understand the lecture concepts through the online sessions as compared to the face-to-face sessions?	Very Difficult	9	14.5	3	4.9	0	0.0	2	4.0	0.003 **
	Difficult	7	11.3	18	29.5	15	27.8	3	6.0	
	Neutral	32	51.6	26	42.6	22	40.7	28	56.0	
	Easy	10	16.1	8	13.1	14	25.9	13	26.0	
	Very easy	4	6.5	6	9.8	3	5.6	4	8.0	
	Mean ± SD	2.89 ± 1.06		2.93 ± 1.01		3.09 ± 0.87		3.28 ± 0.86		0.138 ^NS^
Do you agree that the visibility of the faculty members affects the understanding of the lectures?	Strongly agree	11	17.7	7	11.5	5	9.3	5	10.0	0.175 ^NS^
	Agree	15	24.2	22	36.1	19	35.2	23	46.0	
	Neutral	23	37.1	27	44.3	21	38.9	18	36.0	
	Disagree	9	14.5	3	4.9	9	16.7	3	6.0	
	Strongly disagree	4	6.5	2	3.3	0	0.0	1	2.0	
	Mean ± SD	2.68 ± 1.13		2.52 ± 0.89		2.63 ± 0.87		2.44 ± 0.84		0.554 ^NS^
How well did the faculty members explain the topic on Blackboard as compared to face-to-face sessions?	Very good	15	24.2	4	6.6	3	5.6	9	18.0	0.067 ^NS^
	Good	15	24.2	19	31.1	21	38.9	17	34.0	
	Neutral	27	43.5	27	44.3	23	42.6	21	42.0	
	Bad	5	8.1	9	14.8	7	13.0	2	4.0	
	Very bad	0	0.0	2	3.3	0	0.0	1	2.0	
	Mean ± SD	2.35 ± 0.94		2.77 ± 0.90		2.63 ± 0.78		2.38 ± 0.90		0.031 *
What do you think about the performance of online sessions in relation to audio quality as compared to face-to-face lectures?	Very good	12	19.4	7	11.5	3	5.6	9	18.0	0.050 *
	Good	10	16.1	16	26.2	15	27.8	11	22.0	
	Neutral	26	41.9	28	45.9	25	46.3	26	52.0	
	Bad	14	22.6	7	11.5	11	20.4	4	8.0	
	Very bad	0	0.0	3	4.9	0	0.0	0	0.0	
	Mean ± SD	2.68 ± 1.04		2.72 ± 0.98		2.81 ± 0.82		2.50 ± 0.89		0.388 ^NS^
What do you think about the performance of online sessions in relation to the speed of content delivery?	Very fast	10	16.1	6	9.8	4	7.4	4	8.0	0.056 ^NS^
	Fast	11	17.7	11	18.0	14	25.9	14	28.0	
	Neutral	38	61.3	31	50.8	29	53.7	30	60.0	
	Slow	3	4.8	9	14.8	7	13.0	1	2.0	
	Very slow	0	0.0	4	6.6	0	0.0	1	2.0	
	Mean ± SD	2.55 ± 0.82		2.90 ± 0.99		2.72 ± 0.79		2.62 ± 0.75		0.121 ^NS^

NS: Not significant, SD: Standard deviation. *: Significant. **: More significant.

**Table 7 ijerph-19-12825-t007:** Distribution of responses related to technical sessions according to gender.

		Male (n = 132)		Female (n = 95)		
Related to Technical Session		No. of Respondents	% of Respondents	No. of Respondents	% of Respondents	*p*-Value
Did you experience any connectivity-related technical problems during the online sessions that affected your understanding of the lecture?	No problems	29	22.0	12	12.6	0.022 *
	Minimum problems	39	29.5	46	48.4	
	Moderate problems	53	40.2	32	33.7	
	Severe problems	7	5.3	5	5.3	
	Very severe problems	4	3.0	0	0.0	
	Mean ± SD	2.38 ± 0.98		2.31 ± 0.76		0.603 ^NS^
How convenient were the timings of the online sessions you attended?	Very convenient	31	23.5	2	2.1	0.001 ***
	Convenient	21	15.9	32	33.7	
	Neutral	65	49.2	55	57.9	
	Not Convenient	8	6.1	6	6.3	
	Discomforting	7	5.3	0	0.0	
	Mean ± SD	2.54 ± 1.08		2.68 ± 0.62		0.237 ^NS^
How beneficial did you find the online sessions, especially concerning the effects on your remembering of the concepts of the lecture?	Very beneficial	24	18.2	17	17.9	0.741 ^NS^
	Somewhat beneficial	38	28.8	28	29.5	
	Neutral	48	36.4	40	42.1	
	Not Beneficial	18	13.6	8	8.4	
	Not at all beneficial	4	3.0	2	2.1	
	Mean ± SD	2.54 ± 1.04		2.47 ± 0.95		0.595 ^NS^
How comfortable was your learning experience with online sessions from your home as compared to face-to-face sessions?	Very comfortable and able to concentrate better	47	35.6	41	43.2	0.398 ^NS^
	Comfortable and not able to concentrate better	52	39.4	36	37.9	
	Not very comfortable but able to concentrate	19	14.4	11	11.6	
	Not comfortable and not able to concentrate better	10	7.6	7	7.4	
	Very uncomfortable and a waste of time	4	3.0	0	0.0	
	Mean ± SD	2.03 ± 1.04		1.83 ± 0.91		0.136 ^NS^
Please state your degree of support in favor of such online lecture sessions in the future?	Strongly supported	31	23.5	13	13.7	0.100 ^NS^
	Supported	39	29.5	40	42.1	
	Neutral	47	35.6	36	37.9	
	Unsupported	13	9.8	4	4.2	
	Strongly unsupported	2	1.5	2	2.1	
	Mean ± SD	2.36 ± 0.99		2.39 ± 0.85		0.838 ^NS^

NS: Not significant, SD: Standard deviation. *: Significant. ***: Very significant.

**Table 8 ijerph-19-12825-t008:** Distribution of responses related to understanding of the topic/delivery of the topic according to gender.

		Male (n = 132)		Female (n = 95)		
Related to Understanding of the Topic/Delivery of the Topic		No. of Respondents	% of Respondents	No. of Respondents	% of Respondents	*p*-Value
Did you feel and agree the sessions need to be more interactive?	Strongly agree	31	23.5	21	22.1	0.961 ^NS^
	Agree	54	40.9	41	43.2	
	Neutral	39	29.5	26	27.4	
	Disagree	6	4.5	6	6.3	
	Strongly disagree	2	1.5	1	1.1	
	Mean ± SD	2.19 ± 0.90		2.21 ± 0.89		0.911 ^NS^
How easy do you think it could be to clarify your doubts as related to the lecture topic through Blackboard collaborate?	Very easy	33	25.0	12	12.6	0.008 **
	Easy	30	22.7	34	35.8	
	Neutral	47	35.6	42	44.2	
	Difficult	20	15.2	5	5.3	
	Very Difficult	2	1.5	2	2.1	
	Mean ± SD	2.45 ± 1.07		2.48 ± 0.86		0.824 ^NS^
How easy/difficult was it to understand the lecture concepts through the online sessions as compared to the face-to-face sessions?	Very Difficult	12	9.1	2	2.1	0.001 ***
	Difficult	29	22.0	14	14.7	
	Neutral	51	38.6	57	60.0	
	Easy	25	18.9	20	21.1	
	Very easy	15	11.4	2	2.1	
	Mean ± SD	3.01 ± 1.11		3.06 ± 0.73		0.713 ^NS^
Do you agree that the visibility of the faculty members affects the understanding of the lectures?	Strongly agree	15	11.4	13	13.7	0.233 ^NS^
	Agree	47	35.6	32	33.7	
	Neutral	57	43.2	32	33.7	
	Disagree	11	8.3	13	13.7	
	Strongly disagree	2	1.5	5	5.3	
	Mean ± SD	2.53 ± 0.86		2.63 ± 1.05		0.427 ^NS^
How well did the faculty members explain the topic on Blackboard as compared to face-to-face sessions?	Very good	26	19.7	5	5.3	0.033 *
	Good	38	28.8	34	35.8	
	Neutral	52	39.4	46	48.4	
	Bad	14	10.6	9	9.5	
	Very bad	2	1.5	1	1.1	
	Mean ± SD	2.45 ± 0.97		2.65 ± 0.77		0.101 ^NS^
What do you think about the performance of online sessions in relation to audio quality as compared to face-to-face lectures?	Very good	25	18.9	6	6.3	0.012 *
	Good	33	25.0	19	20.0	
	Neutral	52	39.4	53	55.8	
	Bad	19	14.4	17	17.9	
	Very bad	3	2.3	0	0.0	
	Mean ± SD	2.56 ± 1.03		2.85 ± 0.78		0.021 *
What do you think about the performance of online sessions in relation to the speed of content delivery?	Very fast	18	13.6	6	6.3	0.043 *
	Fast	29	22.0	21	22.1	
	Neutral	66	50.0	62	65.3	
	Slow	14	10.6	6	6.3	
	Very slow	5	3.8	0	0.0	
	Mean ± SD	2.69 ± 0.97		2.71 ± 0.68		0.819 ^NS^

NS: Not significant, SD: Standard deviation. *: Significant. **: More significant. ***: Very significant.

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
