# Peer review of "Perception of Synchronized Online Teaching Using Blackboard Collaborate among Undergraduate Dental Students in Saudi Arabia"

_ijerph, 2022, doi:10.3390/ijerph191912825_

Round 1

Reviewer 1 Report

There is a certain mistake in numbering the reference list. (13. 13. Hagstrom, E, Lindbergb, O. Three theses on teaching and learning in higher education. Teaching in Higher Education, 2013; 276 18(2), 119-128.)

Author Response

Dear Reviewer,

Thank you for your comments. We have benefitted from the wealth of your knowledge and it has enriched our manuscript. The recommended corrections are made and highlighted in GREEN in the attached revised manuscript.

Thank You.

The Authors.

Reviewer 2 Report

The article presents a research work on perceptions about synchronized online teaching among undergraduate students.

The tittle is generic (‘Perception of Synchronized Online Teaching among Undergraduate Dental Students in Saudi Arabia’). However, the research is the result of experiencing ‘blackboard collaborate’, a narrower approach, as the authors recognize in line 229 (limitations of the study)

Comments and suggestions

Minor issues:

Line 15 to 32 (abstract).- The words in bold  ‘introduction’, ‘aim’, ‘materials and method’, ‘results’ and ‘conclusion’ should be deleted. The abstract should be reworded, e.g., instead of ‘Aim: to assess the perception…’ -> ‘The aim of this research word is to assess the perception…’

Line 27.- an extra space is needed… females.Out of the 227 respondents

Line 35.- instead of ‘Introduction: Advancements in …’ -> ‘1. Introduction’ /enter a new line/ ‘Advancements in …’. The same should be applied to line 80 and the other epigraphs, according to the publisher guidelines (in fact, to the elements that should be deleted from the abstract).

Line 42.- To space between the words.

Line 42.- a space is needed: week[2]. -> week [2]; the same happens in other lines (e.g., 45, 47, 49, 53, ...).

Line 46.- ‘necessity’?; perhaps is better to write the term ‘solution’?

Line 110.- study,227 -> study, 227

Line 115.- e.(See table 1) Most -> e, as seen in Table 1.  / End of  paràgraf / Insert Table 1 / new paragraph -> Most...

The same could be applied in line 120, line 130, line 136, line 140, line 146, line 148, line 155.

Line 156.- From this line on, all the tables are included in a row. Once you cite in the text ‘Table i’, you should include the table (Table i) when the paragraph is finished (immediately below the paragraph).

Line 161, Table 4.- An extra space is needed - > your doubtsas related

Line 174.- Schools including dental schools ... - > Educational institutions, including dental schools...

Line 231 to 242.- Author should be refenced by their initials (name & surname).

Line 247: The line should be deleted.

Some considerations …

Materials and Methods (section)

Are there any criteria that you have considered to select the question that you have included in the survey? It should be great to include this explanation in this section.

Discussion (section)

A detailed review of the different results presented in the different tables would be great. Comparing and analyzing the different results showed in each single table about each single topic that is captured in each table would enrich this section.

Conclusion (section)

This section could include in a synthetic way which one is the aim of the research in line with what is written in line 77 (‘perception of online teaching and learning as well as to find out areas that might appear as barriers to quality online teaching and learning’), besides de conclusions that the authors have found once analyzed the results in line with the aim of the research work.

In the abstract, two main ideas are showed as conclusions: technical problems when talking about connectivity & support to the continuation of online lectures.

In the conclusion section it can be read: ‘positive perception of our respondents to online lectures using blackboard collaborate, however, efforts should be made to make them more interactive through means such as gamification and also by allowing the students to view the lecturer during the sessions. Furthermore, online lectures should not be longer than 45mins and minimum of 15mins interval should be allowed between lectures.

So, both contents (abstract & conclusion section) should be harmonized in order to get consistency,

Conclusions of the research should be presented in this section, according to the results that you have obtained from the survey… But - > For instance, it is mentioned that: ‘efforts should be made to make them more interactive through means such as gamification’) is not researched in the survey’. Is this a conclusion of the research? What suports this idea, besides the references to other research Works presented in line 210 to 212? In the survey there is no question about gamification.

References

Including some additional references should be great to improve the introduction. Specifically, references about other online practices introduced as a consequence of the COVID-19 pandemic (i.e. smart learning deployments…)

Several DOI are missed… e.g.:

Exploring Two Teacher Education Online Learning Designs A Classroom of One or Many?

Priscilla Norton & Dawn Hathaway

Journal of Research on Technology in Education; Volume 40, 2008 - Issue 4

https://doi.org/10.1080/15391523.2008.10782517

-

Davis, Julie M. Aspro; Lennox, Sandra; Walker, Sue; and Walsh, Kerryann (2007) "Exploring Staff Perceptions: Early Childhood Teacher Educators Examine Online Teaching and Learning Challenges and Dilemmas,"

International Journal for the Scholarship of Teaching and Learning: Vol. 1: No. 2, Article 8.

https://doi.org/10.20429/ijsotl.2007.010208

-

Line 275 & line 279.- there are some words in bold

...

-

Author Response

Dear Reviewer,

Thank you for your comments. We have benefitted greatly from the wealth of your knowledge and it has enriched our manuscript. The recommended corrections are made and highlighted in YELLOW in the attached revised manuscript.

Thank You.

The Authors.

Reviewer 3 Report

The article is well structured and succeeds in answering the stated objectives. I recommend some suggestions for improvement:

-Increase the bibliographical references, preferably more from the last 5 years.

-Revise the references, the first one has the year in brackets, bold letters appear, etc. 

-I miss in the conclusions the future lines of research that can be derived from this work.

It is a current work, of interest to today's teachers who need to adapt to new teaching methods and the needs of their students in a world characterised by digitalisation. For this reason it is relevant to the educational community.

    After COVID 19, students and teachers have become aware of the possibilities of virtual methods, so a new way of teaching has opened up, which should be well known and the advantages it brings to education should be studied. This is why this work is of scientific interest.

Author Response

Dear Reviewer,

Thank you for your comments. We have benefitted from the wealth of your knowledge and it has enriched our manuscript. The recommended corrections are made and highlighted in GREY in the attached revised manuscript.

Thank You.

The Authors.
